# Elemental Sulfur Inhibits Yeast Growth via Producing Toxic Sulfide and Causing Disulfide Stress

**DOI:** 10.3390/antiox11030576

**Published:** 2022-03-17

**Authors:** Tianqi Wang, Yuqing Yang, Menghui Liu, Honglei Liu, Huaiwei Liu, Yongzhen Xia, Luying Xun

**Affiliations:** 1State Key Laboratory of Microbial Technology, Shandong University, 72 Binhai Road, Qingdao 266237, China; wangtq777@mail.sdu.edu.cn (T.W.); yangyuq@mail.sdu.edu.cn (Y.Y.); 202112545@mail.sdu.edu.cn (M.L.); lhl@sdu.edu.cn (H.L.); liuhuaiwei@sdu.edu.cn (H.L.); 2School of Molecular Biosciences, Washington State University, Pullman, WA 991647520, USA

**Keywords:** elemental sulfur, sulfide, GSSG, disulfide stress, ROS, *Saccharomyces cerevisiae*

## Abstract

Elemental sulfur is a common fungicide, but its inhibition mechanism is unclear. Here, we investigated the effects of elemental sulfur on the single-celled fungus *Saccharomyces cerevisiae* and showed that the inhibition was due to its function as a strong oxidant. It rapidly entered *S. cerevisiae*. Inside the cytoplasm, it reacted with glutathione to generate glutathione persulfide that then reacted with another glutathione to produce H_2_S and glutathione disulfide. H_2_S reversibly inhibited the oxygen consumption by the mitochondrial electron transport chain, and the accumulation of glutathione disulfide caused disulfide stress and increased reactive oxygen species in *S. cerevisiae*. Elemental sulfur inhibited the growth of *S. cerevisiae*; however, it did not kill the yeast for up to 2 h exposure. The combined action of elemental sulfur and hosts’ immune responses may lead to the demise of fungal pathogens.

## 1. Introduction

Sulfur (S) is the fifth most common element on earth, and elemental sulfur mainly exists as octasulfur (S_8_) [1]. Elemental sulfur is hardly soluble in water but can be dissolved in organic solvents, such as methanol and acetone [2]. It is widely used as a fungicide to treat plant diseases [3,4] and fungal infections in animals and humans [5,6,7,8]. Although its fungicidal mechanism has been extensively investigated, the consensus has not been reached [9]. An early theory indicated that elemental sulfur is reduced by fungi to produce H_2_S, which is toxic to cells [10,11,12]. The theory was challenged when potassium permanganate, which immediately oxidizes H_2_S, did not affect elemental sulfur inhibition to fungi [13,14]. Another theory is that elemental sulfur can enter fungal cells and spores and in the cytoplasm, where it directly inhibits the electron transport chain for respiration via modifying important protein thiols [15,16]. Sato et al. recently reported that glutathione and glutathione reductase is involved in reducing elemental sulfur to H_2_S in fungi, and the inactivation of glutathione reductase makes the mutant more sensitive to elemental sulfur [17], and the finding supports the latter theory. A reinvestigation may clarify the toxic mechanism of elemental sulfur.

It is unclear how elemental sulfur enters the cells, but it is likely to present as soluble inorganic and organic polysulfide (HS_n_^–^ and RS_n_^−^, n > 2) in the cytoplasm [15,16]. The insoluble elemental sulfur may first become HS_n_^−^, which then enters the cell [17]. Since glutathione (GSH) is the main low molecular weight (LMW) thiol in plants, animals, fungi, and most bacteria [18], it can reduce elemental sulfur to H_2_S. We recently reported how HS_n_^−^ and GS_n_^−^ are converted to S_8_ in the bacterial cytoplasm [19]. A recombinant *Escherichia coli* strain with cloned sulfide:quinone oxidoreductase oxidizes H_2_S to HS_n_^−^ and glutathione polysulfide (GS_n_^−^) in the cytoplasm, where GSH rapidly reacts with HS_n_^−^ and GS_n_^−^ to produce glutathione disulfide (GSSG) and H_2_S. When intracellular GSH is depleted to low levels, HS_n_^−^ elongates to HS_9_^−^ before forming S_8_, which is the common form of elemental sulfur. This observation promoted us to investigate whether the elemental sulfur toxicity against fungi was due to its rapid reaction with GSH to produce H_2_S and GSSG, in which elemental sulfur acts as an oxidant.

Elemental sulfur, HS_n_^−^, and RS_n_^−^ are also known as sulfane sulfur, as they either contain or can generate zero-valent sulfur [20]. HS_n_^−^ and RS_n_^−^ may function as antioxidants because they are more nucleophilic than small thiols for scavenging reactive oxygen species (ROS) [21]. Here, they function as reductants. Their most important antioxidant activity is to scavenge hydroxyl radical due to the much faster reaction rate than that between DNA and the radical [22]. Sulfane sulfur also displays electrophilic properties, as it can transfer the electrophilic zero-valent sulfur to protein thiols to generate protein-SSH, which may affect certain enzyme activities or functions [23,24]. Since protein persulfidation may protect protein thiols from irreversible oxidation, this is also considered an antioxidant property [25]. However, the similarity in chemical properties between H_2_O_2_ and H_2_S_2_ (or HS_n_^−^) has promoted DeLeon et al. to propose that sulfane sulfur species may function as oxidants [26]. Thus, sulfane sulfur possesses both reductant and oxidant properties.

Here, we report that elemental sulfur acted as a strong oxidant to inhibit the growth of *Saccharomyces cerevisiae*, as it quickly entered the cytoplasm and reacted with GSH to produce H_2_S and GSSG. H_2_S further inhibited the electron transport chain and glucose oxidation, and GSSG induced disulfide stress and oxidative stress to cells. However, elemental sulfur did not kill the cells of *S. cerevisiae*, and the inhibition effect is reversible. Our results explained the mechanisms of elemental sulfur toxicity against *S. cerevisiae*, which offers insights into its action against pathogenic fungi.

## 2. Materials and Methods

### 2.1. Bacterial Strains, Culture Conditions, and Reagents

*Escherichia coli* MG1655 (wild type) and *Staphylococcus aureus* ATCC 6538 (wild type) were grown in lysogeny broth (LB) medium (0.5% yeast extract, 1% peptone, and 1% NaCl) at 37 °C with shaking. *Saccharomyces cerevisiae* BY4742 (MATα his3∆1 leu2∆0 lys2∆0 ura3∆0) was grown in yeast extract-peptone-dextrose (YPD) medium (1% yeast extract, 2% peptone, and 2% glucose) at 30 °C with shaking. Sodium hydrosulfide (NaHS, H_2_S donor) and sublimed sulfur powder were purchased from Sigma-Aldrich (St. Louis, MO, USA). The elemental sulfur stock solution was prepared by dissolving sublimed sulfur powder in acetone.

### 2.2. Preparation of Resting Cells

The selected microorganisms were cultured in a 50 mL medium and grown with shaking. Cells were then centrifuged at 5000× *g* for 10 min. The harvested cells were gently washed once and suspended in 100 mM phosphate buffer saline (PBS) buffer (pH = 7.4) to get the resting cells. When ZnCl_2_ was used, the resting cells were washed and resuspended in 50 mM Tris-HCl buffer (pH = 7.4), and 200 or 500 μM ZnCl_2_ was added to the cell suspension.

### 2.3. Measurement of Growth Curves

Fresh cells of *E. coli*, *S. aureus*, *S. cerevisiae* were inoculated in 5 mL medium and grown overnight at suitable temperature with shaking at 200 rpm. The cells with the initial OD_600nm_ at 0.05 were transferred into 400 µL fresh medium in a 48-well plate. If necessary, elemental sulfur at defined concentrations was mixed with the cultures. Growth curves were plotted by measured OD_600nm_ after incubating the plate in a microplate reader (Synergy H1, BioTek, Winooski, VT, USA) at suitable temperatures with shaking.

To examine the growth inhibition effect of GSSG on *S. cerevisiae*, the resting cells at OD_600nm_ = 2.0 were incubated with 200 μM elemental sulfur solved in acetone or acetone alone at 30 °C for 30 min before they were washed and suspended in 100 mM PBS. These cells were transferred into 400 µL YPD medium with initial OD_600nm_ at 0.05 in a 48-well plate at 30 °C with shaking. The growth curves were measured by using the microplate reader.

### 2.4. Minimal Inhibit Concentration (MIC) Assay

*E. coli*, *S. aureus*, *S. cerevisiae* were cultured to OD_600nm_ of 0.5. Equal amounts of cells (OD_600nm_ = 0.05) were transferred into 400 µL of the appropriate medium containing different concentrations of elemental sulfur in 24-well plates. The 24-well plates were incubated at the proper temperature for 12 h to observe and determine MIC.

### 2.5. Survival Assay

The resting cells of *S. cerevisiae* were prepared and aliquot with OD_600nm_ of 2.0. Indicated concentrations of elemental sulfur were incubated with the cells. Then, the cells were incubated at 30 °C with shaking. At defined time intervals, cells were taken, diluted, and spread onto YPD solid plates (YPD medium with 2.0% agar). The plates were incubated for 48 h before counting. Three parallel experiments were performed to get STDEV, and the data could be used for statistical analysis.

### 2.6. The Detection of H_2_S and Elemental Sulfur

For H_2_S analysis, the sample was centrifuged at 13,000× *g* for 2 min, and 50 μL of the supernatant was derivatized with monobromobimane (mBBr) and detected by high-performance liquid chromatography (HPLC), according to a published method [27]. A reported method was used for elemental sulfur analysis [28,29]. In brief, the sample was directly derivatized with methyl trifluoromethanesulfonate in methanol and was detected by using HPLC with UV detection.

### 2.7. Measuring Oxygen Consumption

Resting cells at OD_600nm_ of 2.0 were incubated with defined concentrations of H_2_S or elemental sulfur at 30 °C. Glucose was immediately added to a final concentration of 2%. The dissolved oxygen was monitored by using an Orion RDO meter (Thermo Scientific Inc., Waltham, WA, USA). Before use, the RDO meter was calibrated with air-saturated water according to the manufacturers’ instructions. The resting cells without mixing with H_2_ or elemental sulfur were used as control.

### 2.8. GSH and GSSG Detection

The *S. cerevisiae* resting cells at an OD_600nm_ of 2.0 were incubated with elemental sulfur at 30 °C for 30 min. A total of 25 mL of resting cells were centrifuged, washed, and suspended in 2 mL of a stock buffer (143 mM sodium phosphate and 6.3 mM Na_4_EDTA, pH = 7.4). Samples were added with 0.1 mL of 10% 5-sulfosalicylic acid [30] and broken through a pressure cell (SPCH-18, Stansted Fluid Power Ltd., Harlow, UK). GSH and GSSG in the cell lysate were assayed by using a kit (Beyotime, Shanghai, China).

### 2.9. ROS Determination

Cellular ROS was measured using the DCFH-DA probe (Beyotime, Shanghai, China). The resting cells of *S. cerevisiae* at an OD_600nm_ of 2.0 were incubated with 10 μM DCFH-DA at 37 °C for 30 min in the dark and were washed three times with 100 mM PBS (pH = 7.4) to remove the extra DCFH-DA. Indicated concentrations of elemental sulfur or H_2_S were added into the cells. The reaction mixtures were incubated at 37 °C for 1 h in the dark. Then, the fluorescence of the cells was measured using a microplate reader (BioTek, Synergy H1) with the excitation wavelength at 488 nm and the emission wavelength at 525 nm.

## 3. Results

### 3.1. Elemental Sulfur Inhibits the Growth of S. cerevisiae

The effect of elemental sulfur on *E. coli*, *S. aureus*, and *S. cerevisiae* was tested. *E. coli* was the least affected (Figure 1A). It could grow even with 1 mM elemental sulfur, but the final OD_600nm_ was only one-quarter of that without elemental sulfur. As elemental sulfur was dissolved in acetone, the same volume of acetone used to dissolve elemental sulfur was also tested, and the volume of acetone used to deliver 1 mM elemental sulfur also inhibited the final OD_600nm_ of *E. coli*, but the inhibition by acetone on *S. aureus* and *S. cerevisiae* did not affect the final OD_600nm_ (Appendix A). The reduced growth yield for *E. coli* was likely due to the effect of acetone. *S. aureus* and *S. cerevisiae* were completely inhibited by 1 mM elemental sulfur (Figure 1B,C). With 500 μM elemental sulfur, both *S. aureus* and *S. cerevisiae* grew as similar as the control groups without elemental sulfur. One apparent effect is that the log phase of cell growth was prolonged (Figure 1B,C). Elemental sulfur has a more serious inhibitory effect on yeast growth than these two bacteria species (Figure 1). Second, the MIC of elemental sulfur to cells was assessed. The MIC values for *S. cerevisiae* and *S. aureus* were 200 and 1000 μM, respectively (Appendix A). Because the growth of *E. coli* was affected by acetone (Appendix A), the MIC value for *E. coli* was not measured. Hence, the MIC value for yeast is lower than that of bacteria too. Third, the viability of *S. cerevisiae* in the presence of 100, 200, 500, and 1000 μM was detected. Within 2 h exposure, elemental sulfur did not show significantly lethal to *S. cerevisiae* cells (Figure 1D). These results indicate that elemental sulfur inhibits the growth of *S. cerevisiae*, but does not kill the cells.

### 3.2. Elemental Sulfur Enters S. cerevisiae to Produce H_2_S That Inhibits the Electron Transport Chain

When *S. cerevisiae* was incubated with elemental sulfur, it was quickly consumed, and H_2_S was rapidly generated within minutes (Figure 2A,B). Oxygen consumption by the yeast was also inhibited after incubating with 100 μM elemental sulfur for 10 min (Figure 2C). The inhibition was likely due to the produced H_2_S, as the incubation of *S. cerevisiae* with 10 μM H_2_S for 1 or 10 min severely inhibited its oxygen consumption (Figure 2C), while the incubation with 10 μM elemental sulfur for 10 min did not affect its oxygen consumption (Figure 2C). Adding 10 μM elemental sulfur to the resting cells, *S. cerevisiae* produced 2.8 μM H_2_S (Figure 2B), which was not sufficient to inhibit the electron transport chain. Alternatively, *S. cerevisiae* was first incubated with 10 μM H_2_S for 10 min. Then, the cells were collected by centrifugation to remove H_2_S. The oxygen consumption by the resuspended cells was not inhibited (Figure 2D), indicating that the H_2_S inhibition on the oxygen consumption by *S. cerevisiae* is reversible. On the other hand, the oxygen consumption by *E. coli* was not affected by 50 to 200 μM H_2_S or elemental sulfur (Appendix A).

Zn^2+^ could quickly react with H_2_S to form the precipitation of ZnS. Hence, Zn^2+^ was used to confirm that H_2_S was responsible for the inhibition of oxygen consumption by *S. cerevisiae* after incubating with elemental sulfur. Oxygen consumption by *S. cerevisiae* was not inhibited by 200 μM ZnCl_2_ (Figure 3A). When the resting cells of *S. cerevisiae* were incubated with 200 μM ZnCl_2_ and different amounts of H_2_S, oxygen consumption of the resting cells were tested. A total of 50 μM H_2_S did not affect the oxygen consumption, but 100 and 200 μM H_2_S did (Figure 3B). Further, 200 μM ZnCl_2_ completely stopped the inhibition of 200 μM elemental sulfur on oxygen consumption by the yeast (Figure 3A). Collectively, these results confirm that elemental sulfur is reduced to H_2_S, which reversibly inhibits the electron transport chain for oxygen consumption.

### 3.3. Elemental Sulfur Enters S. cerevisiae and Generates Disulfide Stress

After *S. cerevisiae* cells were incubated with 200 μM elemental sulfur for 30 min, the cellular GSH was greatly reduced, and GSSG has increased accordingly (Figure 4A,B). The GSSG fraction of the total glutathione (GSH + GSSG) changed from 12.1% to 90.3% after the incubation (Figure 4C). A total of 100 to 500 μM ZnCl_2_ in the buffer did not stop the oxidation of GSH by using 100 μM elemental sulfur, as the percentage of GSSG in *S. cerevisiae* cells was not affected by ZnCl_2_ (Figure 4D).

The presence of GSSG would cause disulfide stress on cells [31,32]. We also detected intracellular ROS after the incubation of *S. cerevisiae* with elemental sulfur for 1 h. A total of 200 and 500 μM elemental sulfur increased intracellular ROS by 60.2% and 194.9%, respectively (Figure 5A), indicating that the cells were under oxidative stress. When *S. cerevisiae* cells that had accumulated GSSG were inoculated into a fresh YPD medium, its growth rate was only initially inhibited (Figure 5B). It seems that the cells reduced GSSG in vivo and restored redox homeostasis before growing.

## 4. Discussion

Our results indicate that the fungicidal effect of elemental sulfur is due to its function as a strong oxidant (Figure 6). Inside *S. cerevisiae*, it is reduced to H_2_S, which inhibits the electron transport chain, and GSH is oxidized to GSSG creating disulfide stress and oxidative stress [33,34,35]. Elemental sulfur does not directly inhibit the electron transport chain, as ZnCl_2_ stopped the inhibition of aerobic respiration by sequestering the produced H_2_S but not affecting GSSG formation (Figure 4). It is known that fungal cells and spores can rapidly take up elemental sulfur [16]. Because elemental sulfur is insoluble, it has been hypothesized that elemental sulfur is converted to HS_n_^–^ that passes through the cell membrane and enters the cell [17]. Elemental sulfur likely reacts with the cell-released H_2_S to produce HS_n_^−^ [36]. This process is challenged with our results, as Zn^2+^ stopped H_2_S inhibition of the electron transport chain, but not GSSG formation after elemental sulfur exposure (Figure 4). Our results implied that elemental sulfur directly enters the cell and then reacts with GSH to produce H_2_S and GSSG, which is spontaneous and quick [19]. The diffusion of H_2_S into the gas phase may facilitate GSSG formation [37,38]. The mechanism of elemental sulfur uptake warrants further investigation.

The fungicidal effect of elemental sulfur via H_2_S has been proposed [9], as H_2_S is toxic [39,40,41]. The toxicity of H_2_S is mainly through inhibition of cytochrome c oxidase of the electron transport chain [41,42,43]. However, H_2_S alone is not sufficient for the toxicity of elemental sulfur [13,14], and blocking H_2_S production makes yeast cells more sensitive to elemental sulfur [17]. H_2_S inhibition will gradually recover, accompanied by the volatilization and decomposition of H_2_S [44]. Our results also support that the inhibition of H_2_S on the yeast electron transport chain is reversible (Figure 2). The inhibition of elemental sulfur on the aerobic respiration of *E. coli* is minimal (Appendix A), which is consistent with the fact that the bacterium does not use cytochrome c oxidase for aerobic respiration [45]. Thus, H_2_S is partially responsible for the toxic effect of elemental sulfur.

The accumulation of GSSG will lead to disulfide stress [31,32]; GSSG is normally reduced by glutathione reductase back to GSH [46]. When a large amount of elemental sulfur enters the cell, glutathione reductase fails to reduce the rapidly produced GSSG back to GSH, leading to GSSG accumulation (Figure 4). This phenomenon is consistent with the result in our previous study [19]. The accumulation of GSSG induces disulfide stress [17], which is accompanied by increased ROS in *S. cerevisiae* after exposure to elemental sulfur (Figure 5). The increased ROS is likely due to the lack of GSH to remove ROS [33]. The accumulation of GSSG is an indication of protein thiol oxidation and modification, which affect enzyme activities and gene regulation [21]. When H_2_S is removed, the yeast cells resume O_2_ consumption (Figure 2). The cells with GSSG accumulation displayed a delayed growth (Figure 5), indicating that the cells need to reduce GSSG before resuming normal growth. Sulfide has been reported to affect oxidative stress [47,48], but we did not detect the change of reactive oxygen species in *S. cerevisiae* after exposing it to 200 μM H_2_S (Figure 5).

Oxidative stress refers to a state in which the oxidant and antioxidant effects in the cell are out of balance with more oxidants [49]. After cells are subjected to oxidative stress, more peroxides and free radicals are produced. These ROS damage cell proteins, lipids, and DNA [49,50]. Exposure to elemental sulfur results in GSH oxidation, and GSH deficiency leads to increased ROS and oxidative stress [33]. This can explain why other studies have shown that elemental sulfur can damage cell structure and proteins [51]. ROS may also cause apoptosis of the yeast [52], but we did not see severe cell death after the cells were exposed to elemental sulfur.

Elemental sulfur in the form of octasulfur is the most stable form of zero-valent sulfur, which could act both as nucleophile and electrophile [53]. When elemental sulfur enters the cytoplasm, it reacts with cellular thiols to become HS_n_^−^ and GS_n_^−^ [19], both of which are more reactive than thiols as the reductants [21]. The electrophilic property of HS_n_^−^ and GS_n_^−^ is often emphasized for the transfer of a sulfur atom to protein thiols, which causes protein thiol persulfidation (RSSH) and other modifications, affective enzyme and gene regulator activities, due to signaling and regulating effects [23,54,55,56,57]. Persulfides react with cellular thiols to produce disulfides. Under normal physiological conditions, glutathione reductase and other enzymes will reduce disulfides back to thiols. The effect of their function as oxidants is muted in most cases. However, when elemental sulfur is supplied in excess, it leads to disulfide accumulation, its role as an oxidant becomes clear.

Based on previous reports and our findings, we proposed the inhibition mechanism of elemental sulfur (Figure 6). Elemental sulfur enters *S. cerevisiae*, and it reacts with GSH to form GSSH, which will further react with another GSH to form H_2_S and GSSG. The produced H_2_S inhibits the electron transport chain (Figure 2). The inhibition of the electron transport chain decreases ATP synthesis and glucose metabolism [58]. The accumulation of GSSG induces disulfide stress and oxidative stress, which collectively slow down yeast growth (Figure 5). The fungicidal mechanism of elemental sulfur is mainly through the produced H_2_S and GSSG, which will slow down the growth of the yeast. Hence, the toxic effect of elemental sulfur is due to its function as an oxidant.

The toxic mechanism toward the yeast should also apply to pathogenic fungi. If elemental sulfur is used as a fungicide, its inhibition of pathogenic fungi could provide the immune systems of plants and animals a chance to destroy them. Further, elemental sulfur is often used to protect grapevines, raspberries, and blackberries against parasitic mites, such as *Calepitrimerus vitis*, *Colomerus vitis*, and *Acalitus orthomera* [59,60]. Our findings may also apply to the toxicity against mites. These findings extended our understanding of the toxic mechanism of elemental sulfur, which is instructive for the treatment of fungal infections in plants and animals and the protection of plants against pathogenic mites.

## Figures and Tables

**Figure 1 antioxidants-11-00576-f001:**
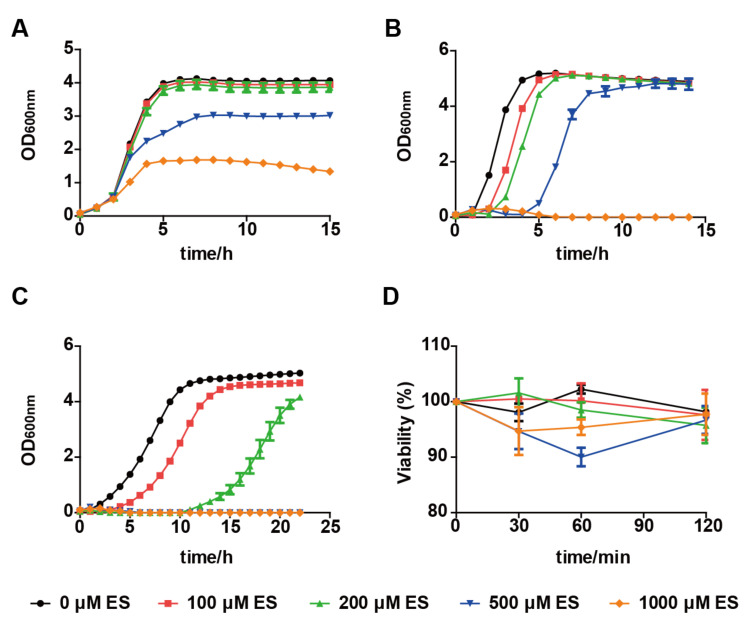
Characteristics of the response of *E. coli*, *S. aureus,* and *S. cerevisiae* to elemental sulfur. The elemental sulfur inhibited the growth of *E. coli* (**A**), *S. aureus* (**B**), and *S. cerevisiae* (**C**). Elemental sulfur was added to the cell cultures at the concentrations as indicated in the figure. (**D**) Survival assay for *S. cerevisiae*. Indicated concentrations of elemental sulfur were added to the resting cells of *S. cerevisiae* at an OD_600nm_ of 2.0. After 30, 60, and 120 min, samples were diluted and plated onto YPD plates, respectively. Colonies counting was performed after 48 h incubation at 30 °C. Data are averages of three parallel experiments with standard deviations. ES, elemental sulfur.

**Figure 2 antioxidants-11-00576-f002:**
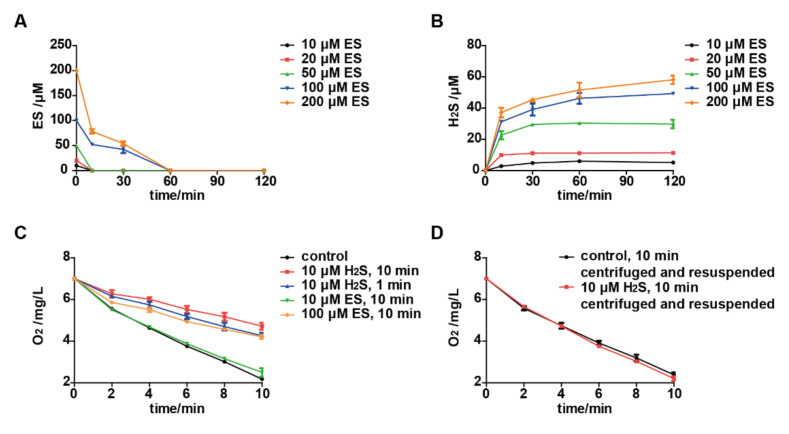
H_2_S inhibits the electron transport chain of *S. cerevisiae*. After adding elemental sulfur to the resting cells of *S. cerevisiae*, the residual concentrations of elemental sulfur (**A**) and H_2_S (**B**) were analyzed at different time intervals. (**C**) Oxygen consumption was measured after incubating 10 and 100 μM elemental sulfur for 10 min or 10 μM H_2_S with the resting cells of *S. cerevisiae* for 1 or 10 min, respectively. (**D**) The resting cells of *S. cerevisiae* were mixed with 10 μM H_2_S first for 10 min. Then, the oxygen consumption was measured after H_2_S was removed. The resting cells of *S. cerevisiae* used were OD_600nm_ = 2.0. The 2% glucose was added as the substrate of the electron transport chain. Data are averages of three parallel experiments with standard deviations. ES, elemental sulfur.

**Figure 3 antioxidants-11-00576-f003:**
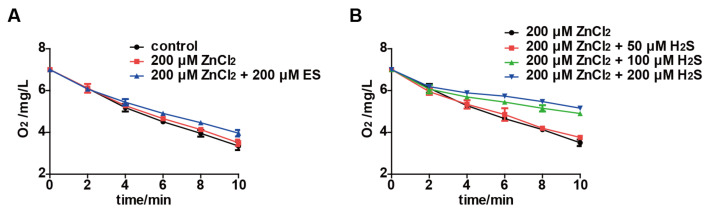
Zn^2+^ prevents H_2_S from inhibiting yeast from consuming O_2_. (**A**) Oxygen consumption was determined after incubating 200 μM ZnCl_2_ with *S. cerevisiae* resting cells in the presence or absence of 200 μM elemental sulfur for 10 min. (**B**) Oxygen consumption was determined after *S. cerevisiae* resting cells were incubated with different concentrations of H_2_S for 10 min in the presence of 200 μM ZnCl_2_. The *S. cerevisiae* resting cells used in the study were OD_600nm_ = 2.0. The 2% glucose was added as the fuel of the electron transport chain. Data are averages of three parallel experiments with standard deviations. ES, elemental sulfur.

**Figure 4 antioxidants-11-00576-f004:**
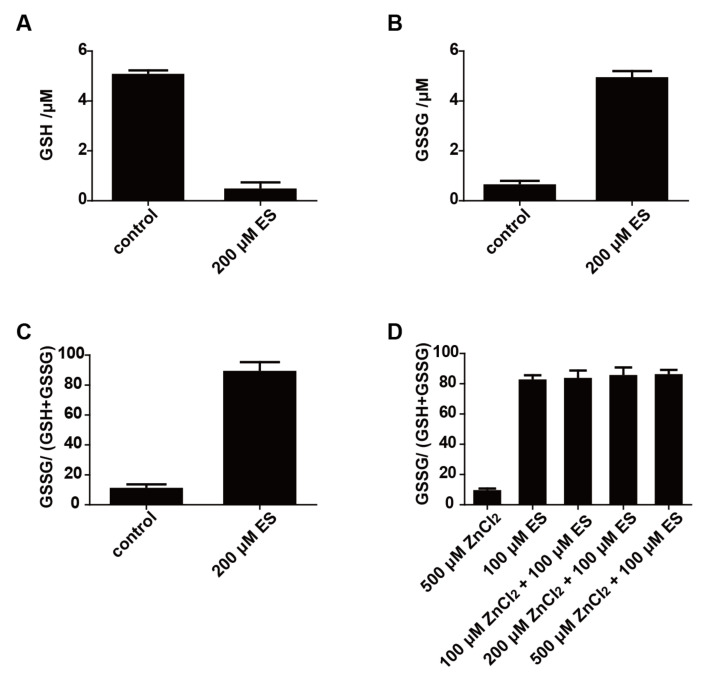
The effect of elemental sulfur on reduced and oxidized glutathione. The concentration of (**A**) GSH and (**B**) GSSG were measured in the resting cells of *S. cerevisiae* after incubation with elemental sulfur for 30 min. (**C**) The proportion of GSSG is increased after elemental sulfur was incubated with *S. cerevisiae* resting cells. (**D**) The proportion of GSSG was increased after elemental sulfur was incubated with *S. cerevisiae* resting cells in the presence of up to 500 μM ZnCl_2_. The *S. cerevisiae* resting cells used was at an OD_600nm_ of 2.0. Data are averages of three parallel experiments with standard deviations. ES, elemental sulfur.

**Figure 5 antioxidants-11-00576-f005:**
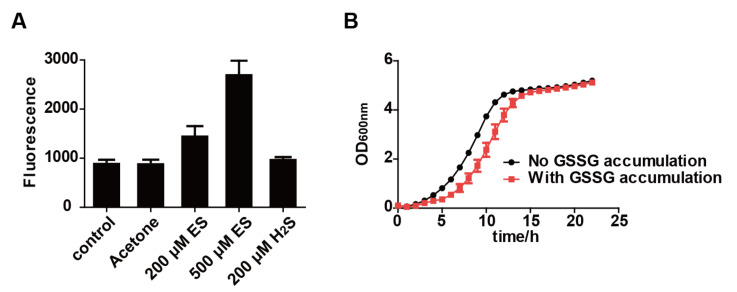
The accumulation of GSSG caused oxidative stress and inhibited cell growth. (**A**) Intracellular ROS of *S. cerevisiae* resting cells with or without GSSG accumulation. The resting cells of *S. cerevisiae* were set at OD_600nm_ = 2.0. (**B**) The growth of *S. cerevisiae* with or without GSSG accumulation. The initial OD_600nm_ was adjusted to 0.05. Data are averages of three parallel experiments with standard deviations. ES, elemental sulfur.

**Figure 6 antioxidants-11-00576-f006:**
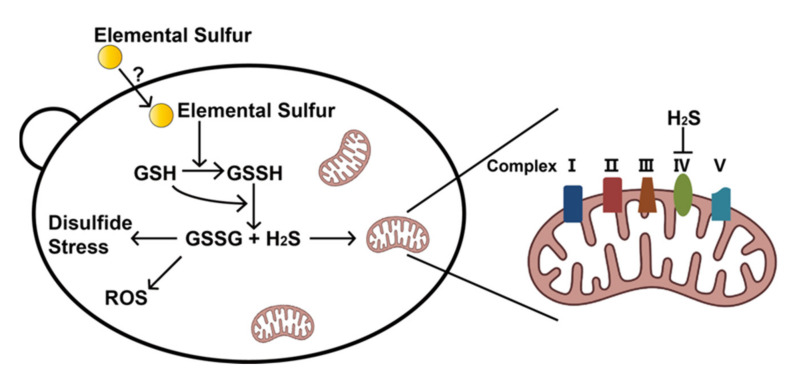
The inhibition mechanisms of elemental sulfur against *S. cerevisiae*.

## Data Availability

Data are contained within the article and its Appendix A.

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
