# Peer review of "Elemental Sulfur Inhibits Yeast Growth via Producing Toxic Sulfide and Causing Disulfide Stress"

_antioxidants, 2022, doi:10.3390/antiox11030576_

Round 1
Reviewer 1 Report
Sulphur is commonly used as a fungicide to protect grapevines against Calepitrimerus vitis and Colomerus vitis, as well as apple trees, peach trees, cucurbits, cereals, tomatoes, carrots, parsley, hops, ornamental plants and sugar beet against Erysiphales, grass seed crops against Septroria, apricots against Monilinia, raspberry and blackberry plants against Eriophyidae or fruit trees and ornamental plants nurseries against Erysiphales.
The paper “Elemental sulfur inhibits yeast growth via producing toxic sulfide and causing disulfide stress” by Tianqi Wang, Yuqing Yang, Menghui Liu, Honglei Liu, Huaiwei Liu, Yongzhen Xia and Luying Xun describes a study of elemental sulphur effect on Saccharomyces cerevisiae cells.
The authors proved that sulphur inhibits the growth of studied yeast cells. Elemental sulphur is quickly consumed by the cells, H2S is rapidly generated and it inhibits the electron transport chain. The cellular thiols, such as GSH, are oxidized to GSSG creating oxidative stress. The authors proposed the inhibition mechanism of elemental sulphur schematics and proved in a series of test using Zn2+ to stop the inhibition that the fungicidal mechanism of elemental sulphur is mainly through the produced H2S and GSSG which slows down the growth of pathogenic fungi, including yeast.
The authors emphasize that elemental sulphur did not kill the yeast for up to 2-hour exposure, and the growth inhibition was reversible. However, the host must be protected from the pathogen especially in time of highest vulnerability; thus, understanding of sulphur uptake and inhibition mechanism could be essential for proper timing and minimum necessary dosing of the fungicide.
The paper is clearly organised, given in good, simple scientific English, methods are well described, charts and figures appropriate. I missed the HPLC parameters a bit, but the reference to standard method is sufficient.
It would have been nice if the authors could apply the fungicide not only to these bacterial strains, but also to species against sulphur fungicides are more often used. Maybe, it might be a good topic for the next paper.
Also, this is the Antioxidants Journal, so the part about oxidative stress could be emphasized a bit more, both in the Introduction and in Discussion.
I believe the paper is interesting for the readers of Antioxidants and I recommend it to be accepted. However, it will be marked as “minor revision” to give the authors a chance to widen the part about oxidative stress, if they decide to do so.
Author Response
Thank you for your encouragements and suggestions. We mentioned that elemental sulufr is often used to protect fruit trees against parasitic mites in the discussion. We also revised the abstract, introduction, and discussion to clearly show that elemental sulfur acts as a strong oxidant for its inhibition to fungi.
Reviewer 2 Report
The manuscript entitled “Elemental sulfur inhibits yeast growth via producing toxic sulfide and causing disulfide stress” describes the growth inhibition mechanisms of elemental sulfur against S. cerevisiae. The manuscript is a description of the fungicidal effect of elemental sulfur. It is scientifically sound and well written, easy to read and understand although the introduction could be broadened (e.g. use as a fungicide to treat plant diseases and fungal infections in animals and humans - please describe some examples). The methods used are well chosen and discussion is well written. It will receive attention and readability because of the proposed fungicidal mechanism of elemental sulfur. I recommend to publish the manuscript.
Author Response
Thank you very much for your effort to review our manuscript and gave positive feedback.